# The first wave of COVID-19 in Israel—Initial analysis of publicly available data

Mark Last ⓘ *

Department of Software and Information Systems Engineering, Ben-Gurion University of the Negev, Beer-Sheva, Israel

* mlast@bgu.ac.il

**Data Availability Statement:** The data underlying the results presented in the study are available from Humanitarian Data Exchange https://data.humdata.org/event/covid-19. COVID-19 Data Repository of the Israeli Ministry of Health https://data.gov.il/dataset/covid-19.

## Abstract

The first case of COVID-19 was confirmed in Israel on February 21, 2020. Within approximately 30 days, the total number of confirmed cases climbed up to 1, 000, accompanied by a doubling period of less than 3 days. About one week later, after this number exceeded 4, 000 cases, and following some extreme lockdown measures taken by the Israeli government, the daily infection rate started a sharp decrease from the peak value of 1, 131 down to slightly more than 100 new confirmed cases on April 30. Motivated by this encouraging data, similar to the trends observed in many other countries, along with the growing economic pressures, the Israeli government has quickly lifted most of its emergency regulations. Throughout May, the daily number of new cases stayed at a very low level of 20–40 until at the end of May it started a steady increase, exceeding 1, 000 by the end of June and 2, 000 on July 22. As suggested by some experts and popular media, this disturbing trend may be even a part of a "second wave". This article attempts to analyze the data available on Israel at the end of July 2020, compared to three European countries (Greece, Italy, and Sweden), in order to understand the local dynamics of COVID-19, assess the effect of the implemented intervention measures, and discuss some plausible scenarios for the foreseeable future.

## Introduction

The first case of COVID-19 was confirmed in Israel on February 21, 2020. On March 9, a two-week self-isolation was imposed on all people coming from abroad and on March 12, schools and universities were closed by the government order, partially switching to distant teaching. Then, on March 19, when the number of daily confirmed cases exceeded 100, all non-essential businesses were ordered to close, the employees were required to increase social distancing of their workers and, if possible, allow them to work from home, whereas people's movement outside their homes was restricted significantly. Still, by March 23, the number of confirmed cases climbed up to 1, 000, accompanied by a doubling period of less than 3 days. A continuous decrease in the growth rate has started only around March 29, after the total number of confirmed cases exceeded 4, 000.

**Funding:** The author received no specific funding for this work.

**Competing interests:** The author have declared that no competing interests exist.

With the peak unemployment rate of 24%, mostly due to people placed on unpaid leave as a result of the pandemic, the Israeli government decided to partially lift the lockdown, which has been maintained throughout the Passover holiday (April 7-16). The "exit strategy" steps under consideration included increasing the allowed percentage of workers at workplaces, opening some shops, partially resuming public transportation, gradually getting children back to school, and selectively releasing people from the movement restrictions based on the level of infection in their neighborhood and their risk group. A list of detailed measures for easing the restrictions was approved by the Israeli government on May 04, 2020. By April 30, the doubling period went up to several weeks with only 222 death outcomes out of 15, 946 confirmed cases. However, at the end of May, a few weeks after the government lifted most of its restrictions, the daily number of new cases started a new and prolonged increase, approaching 2, 000 around July 22.

Table 1 attempts to compare COVID-19 evolution in Israel and three European countries: Sweden, Greece, and Italy. Each of these countries has some common characteristics with Israel along with some prominent differences. In terms of population size, Sweden and Greece are similar to Israel, whereas Italy is several times bigger. On the other hand, the median age in Israel is much lower than in the other three countries, indicating a higher percentage of young population. The weather conditions in Israel are quite similar to Greece, slightly warmer than in Northern Italy, and radically different from Sweden. Also, Sweden, as a typical Scandinavian country, is normally characterised by a greater social distancing than such Mediterranean countries as Italy, Greece, and Israel.

Huynh [1] examined the role of the cultural dimension in practising social distancing in the beginning of the COVID-19 epidemic across 58 different countries. Social distancing data was collected from the Google COVID-19 community mobility reports covering the period from 16 February to 29 March 2020. These country–level observations were matched with cultural indices calculated in [2]. The main finding was that the country with higher 'Uncertainty Avoidance Index (UAI)' had less proportion in gathering at public areas during the study period.

The COVID–19 risk perception by the population may have a direct impact on the effectiveness of the public health policy. From a sample of 391 Vietnamese respondents aged from 15 to 47 years, the study [3] found that an extensive use of social media leads to a higher risk perception. In addition, people from central and southern Vietnam regions, which are popular destinations for Chinese tourists, expressed a higher risk perception of COVID-19 than those from northern Vietnam. The survey was conducted in February 2020, during the first three weeks after the official announcement of the epidemic by the Vietnamese government. The same survey also found that the high COVID–19 risk perception caused the Vietnamese citizens to wear masks in public places, even before it became an official government regulation

**Table 1. Comparison of COVID-19 in four countries.**

|  | Israel | Sweden | Greece | Italy |
|---|---|---|---|---|
| Population (M) | 8.655 | 10.099 | 10.423 | 60.461 |
| Median Age | 30 | 41 | 46 | 47 |
| Second confirmed case | 26-Feb | 26-Feb | 27-Feb | 7-Feb |
| First 10 confirmed cases | 1-Mar | 29-Feb | 5-Mar | 21-Feb |
| First 100 confirmed cases | 12-Mar | 6-Mar | 13-Mar | 23-Feb |
| First death case | 21-Mar | 11-Mar | 11-Mar | 21-Feb |
| Mortality rate—July 31, 2020 (per 100K in population) | 5.92 | 57.09 | 1.98 | 58.12 |
| Confirmed cases—July 31, 2020 (per 100K in population) | 819.99 | 763.20 | 42.95 | 409.42 |

on March 16, 2020 [4]. This could be one of the factors that mitigated the first wave of the epidemic in Vietnam. It is noteworthy that at this time (mid–September 2020), Vietnam still has an extremely low mortality rate of 0.04 per 100, 000 in population [5].

As shown in Table 1, the first multiple COVID-19 cases were detected in Israel, Sweden, and Greece around February 26, about three weeks later than in Italy. In all four countries, the number of confirmed cases has reached 100 within 1.5–2.5 weeks, with a first death case reported 2–3 weeks from the apparent start of the epidemic. However, none of the above mentioned characteristics of each country (population size, climate, age distribution, etc.) can explain such extreme differences in COVID–related mortality rates: nearly 10 times higher in Italy and Sweden than in Israel, and 3 times higher in Israel than in Greece.

According to [2], Greece has the highest Uncertainty Avoidance Index (UAI) index of 112, Sweden has one of the lowest (29), whereas Israel and Italy are ranked close to each other with UAI values of 81 and 75, respectively. The high value of UAI in Greece, possibly associated with a low level of massive gatherings [1], may provide a partial explanation for the extremely low mortality rate observed in Greece vs. the other three countries, but, apparently, UAI has nothing to do with the huge gap in mortality rates between Italy and Israel. It is also noteworthy that by the end of July 2020, Israel slightly surpassed Sweden in the number of confirmed cases per 100k in population, both countries having 2 times higher proportion of cases than Italy and 20 times higher than Greece.

The ongoing discussions of alternative "coronavirus strategies" taken by different governments brings up the following questions:

1. What is the basic reproduction number $R_0$ in Israel vs. other countries and how is it evolving over time?

2. Which measures, if any, have caused the steep decrease in the daily number of new confirmed cases in Israel, which has been observed during April–May 2020, and what could be done to reverse the subsequent growth in the infection rate?

3. What is the true Infection Rate (IR) in Israel, i.e. what is the actual fraction of infected people including those who were not tested for coronavirus during their detectable period?

4. What are the Case Fatality Rate (CFR) and the Infection Fatality Rate (IFR), defined as the ratio of COVID-19 attributed deaths to the number of confirmed cases and the total number of infection cases, respectively? Are they changing over time?

5. Finally, the most significant question: What is the expected evolution of the epidemic in Israel as a function of alternative intervention strategies?

The subsequent sections will focus on each one of these crucial questions. Finally, we will discuss the implications of our data analysis on the prospective decisions to be taken by the government in Israel and other countries.

## Materials and methods

### Data sources

For daily case count data in Israel and other countries, we rely upon the Novel Coronavirus (COVID-19) Cases Data, which is part of Humanitarian Data Exchange (HDX) platform [5]. The data is compiled since 22 January 2020 by the Johns Hopkins University Center for Systems Science and Engineering (JHU CCSE) from various sources such as the World Health Organization (WHO) and European Centre for Disease Prevention and Control (ECDC). The

available fields include Province/State, Country/Region, Last Update, Confirmed, Suspected, Recovered, and Deaths.

The regional mortality data in Italy was downloaded from the website of the Italian Department of Civil Protection [6] on July 23, 2020. The available fields for each date and region include (translated from Italian): hospitalized with symptoms, intensive care, total hospitalized, isolated at home, total positive, total positive variation, new positive, discharged recovered, deceased, suspected cases, screening cases, total cases, swabs, cases tested.

The coronavirus geographical data in Israel was downloaded from the COVID-19 Data Repository of the Israeli Ministry of Health [7] on July 23, 2020. The available fields included: town_code, agas_code, date accumulated_tested, new_tested_on_date, accumulated_cases, new_cases_on_date, accumulated_recoveries, new_recoveries_on_date, accumulated_hospitalized, new_hospitalized_on_date, accumulated_deaths, new_deaths_on_date, town.

## Modeling the infection dynamics

Following [8], we assume that the COVID-19 outbreak can be represented by SIR dynamics, which assumes that at any given point of time, each individual in the population belongs to one of the following three states: (I)nfected, (S)usceptible to infection, or (R)emoved from the transmission process. However, the original SIR model and its variations, like SEIR (susceptible–exposed–infected–recovered) [9], are continuous time models based on a set of differential equations. In contrast, we represent the COVID-19 dynamics by a discrete time model, which is more appropriate for the time series data of daily case counts. Fixing the total size of the population under a discrete time SIR model implies that the sum of the daily changes in the amount of individuals belonging to each one of the above three compartments should be equal to zero:

$$I_t + S_t + H_t = 0 \tag{1}$$

where $I_t$, $S_t$, and $H_t$ stand for the changes in the total amount of infected, susceptible, and healed ("recovered") people, respectively, between the days $t-1$ and $t$. Our discrete time model for the temporal evolution of the number of infected individuals builds upon the basic reproduction number $R_0$, which represents the average number of secondary infections an infected person will cause in a completely susceptible population before he or she is effectively removed from the population as a result of recovery, hospitalization, quarantine, etc. [8]. At the time of a pandemic, the value of $R_0$ may be temporarily reduced by decreasing the amount of social interaction (a measure known as "social distancing"). Additional parameters used by our model include the average duration of the incubation period $L_I$ between the exposure and the onset of clinical symptoms and the generation time $[L_I - max\_T_I; L_I]$ between the primary case and the secondary case, where $max\_T_I$ represents the maximum number of days from the start of the infectious period to the onset of the symptoms. We assume that symptomatic individuals are not infectious anymore as they are immediately isolated from the susceptible population and required to take a COVID-19 test. We also assume that asymptomatic people are never referred to a test and that the social behavior of infected individuals does not change during their infectious period. Given the initial size of the susceptible population $N$, we can define a discrete time model of infection dynamics using Eq 2.

$$I_t = \frac{R_0}{max\_T_I} \sum_{i=t-L_I}^{t-L_I+max\_T_I-1} I_i (1 - \frac{CumI_t}{N}) \tag{2}$$

$$CumI_t = \sum_{j=1}^{i} I_j \tag{3}$$

$$R_{eff} = R_0(1 - \frac{CumI_t}{N}) > 1.0 \tag{4}$$

$$CumI_t^* = N(1 - \frac{1}{R_0}) \tag{5}$$

Eq 4 shows the condition that the number of infected people continues to grow exponentially (effective reproduction number greater than 1.0). From there, using Eq 5 we can extract the herd immunity threshold, which is the minimum cumulative amount of infected individuals $CumI_t^*$ that should stop an epidemic under a fixed value of $R_0$.

Given a time series of estimated daily infections $I_i$, the incubation period $L_I$, the generation time range $[L_I - max\_T_I; L_I]$, and the population size $N$, we can use a stochastic optimization method to find the value of $R_0$, which should reconstruct that series with a minimal average absolute error. The error can be further reduced by splitting the time series into several segments and estimating $R_0$ for each segment separately.

To monitor the evolution of $R_0$ on daily basis, we can take several simplifying assumptions. First, we assume that as long as only a small proportion of a country's population is infected, the number of susceptible individuals $S_t$ remains close to the total population size $N$, making no difference between the basic reproduction number $R_0$ and the effective reproduction number $R_{eff}$. We also assume the generation time to be fixed to the midpoint day of the interval $[L_I - max\_T_I; L_I]$. Applying these assumptions to Eq 2, we can approximate the expected number of new infected cases on day $t$ as:

$$I_t = R_0 I_{t-L_I+max\_T_I/2} \tag{6}$$

Eq (6) is a linear autoregressive process of order $L_I - max\_T_I/2 + 1$ and thus, as long as the above assumptions hold, we can use the least squares method to estimate the value of $R_0$ from the daily counts of new infected cases. Since $R_0$ may be affected by multiple factors, such as the level of social distancing, we can monitor its change over time by calculating the moving slope of the autoregression Eq (6) over a sliding window of $n$ days.

To project the epidemic dynamics into the future, we just need to choose the expected value of $R_0$, initialize the daily amounts of infected people on the last $L_I$ days before the beginning of the projection period, and start calculating the succeeding values of $I_t$ recursively using Eq (2). A similar simulation paradigm was implemented in [10] to evaluate several transmission and intervention scenarios for COVID-19 in the United States over the next five years.

As long as there is no daily testing of the entire population, the exact values of $I_t$ remain unknown. However, in the next sub–section, we explain how one can estimate the actual infection dynamics from the available epidemiological data.

## Estimating the actual infection dynamics from available data

We assume that symptomatic individuals take a COVID-19 test on the day of the symptoms onset (the end point of the incubation period), but it takes $r$ days on average to report a positive test result as a "confirmed case". Given the average incubation period $L_I$ and the average reporting delay $r$, the time series of the daily number of new infection cases $I_t$ is related to the

time series of the daily number of confirmed cases $C_t$ by Eq (7).

$$I_t = p_{t+L_I+r} C_{t+L_I+r} \tag{7}$$

where $p_t$ is the reporting rate on day $t$, which is equal to 1 if there is a full and timely detection of all positive cases, based on daily testing of the entire population, and greater than 1 when some under–reporting takes place.

While the amount of new confirmed COVID-19 cases in various countries, regions and even cities is published on daily basis, its level of under–reporting is unknown and potentially unstable due to inconsistent testing policy, varying reporting time, and other factors and thus we cannot use it as a trustworthy indicator of the epidemic dynamics. On the other hand, the daily number of virus–related deaths $D_t$ is expected to be much more reliable than the testing results and it should not suffer from any significant reporting delay. Assuming a fixed Infection Fatality Rate *IFR*, we can estimate the daily amount of infection cases on day $t$ as a function of the amount of death outcomes on day $t + L_I + d$, where $d$ is the average number of days between the onset of the virus symptoms and the patient death (see Eq 8). Hence, disregarding the true value of *IFR*, we can explore the dynamics of the daily death rate $D_t$ as a time–shifted substitute for the true infection rate $IR_t$ (Eq 9). Given the average reporting delay $r$, we can also estimate the Case Fatality Rate (CFR), which is defined in epidemiology as the proportion of people who die from a specified disease among all individuals diagnosed with the disease over a certain period of time [11]. Since the CFR numerator is restricted to deaths among people included in the denominator, we calculate it as the average ratio between the amount of death outcomes on day $t + d - r$ and the number of confirmed cases on day $t$ (Eq 10).

$$I_t = \frac{D_{t+L_I+d}}{IFR} \tag{8}$$

$$IR_t = \frac{\sum_{i=1}^{t} I_i}{N} \tag{9}$$

$$CFR = \frac{D_{t+d-r}}{C_t} \tag{10}$$

## Results

### The epidemic dynamics

Our analysis of COVID–19 evolution is based on the following timing parameters provided in [12, 13]:

- Mean incubation period $L_I = 5.5$ days

- Minimum time from the start of the infectious period to the onset of the symptoms $min\_T_I = 1$ day

- Maximum time from the start of the infectious period to the onset of the symptoms $max\_T_I = 3$ days

- Mean generation period (average number of days it takes a patient to start infecting others) $L_I - max\_T_I/2 = 4$ days

- Mean time between the onset of the virus symptoms and the patient death $d = 11$ days

- Mean time from infection to death $L_I + d = 16.5$ days

- Mean time from symptoms to detection $r$ = 4.5 days

- Mean time from infection to detection $L_I + r$ = 10 days

- Mean time from detection to death $d - r$ = 6.5 days

Fig 1 shows the overall evolution of the reproduction number in Israel between March 27 and July 5, 2020 along with the 14-day moving average of the daily number of death cases between March 22 and July 18, 2020. On each day, $R_0$ was estimated as a moving slope of a linear autoregressive model over a 14-day sliding window of daily death cases Smoothed with a three-day moving average using an autoregression lag of 4 days, which is equal to the mean generation interval, and assuming that the number of deaths on day $d$ represents the number of new infections on day $d - 16$. The chart also shows the main intervention measures taken by the Israeli government during the same period. Since we did not have enough mortality data to estimate the infection rate before March 27, we could not evaluate the direct effect of the school closing, which occurred around March 12, as well as business activity and movement restrictions imposed on March 19. Further measures, such as mandating masks in public (April 1) and Passover lockdown (April 7) were actually introduced when the value of $R_0$ has already decreased below the critical level of 1.0, a trend which continued throughout the first half of May despite a fast relaxation of most restrictions enforced in March and April. As $R_0$ went below the value of 1.0, the doubling rate (number of days required to double the number of infected people) climbed highly above the 10 days threshold, which was defined in the Government decision from May 04, 2020 as one of the criteria for rolling back the "exit strategy" steps.

After reaching an extremely low level of 0.3 around May 17 and shortly after reopening of all schools in the country (which stayed open until the end of the school year on June 20), $R_0$ started to climb up again in a continuous trend, which included four distinct peaks on May 23 (1.1), June 12 (1.8), June 22 (1.3), and July 2 (1.4). The moving average of the daily mortality rate clearly followed the evolution of $R_0$ with a lag of several weeks, peaking at 9.0 per day on

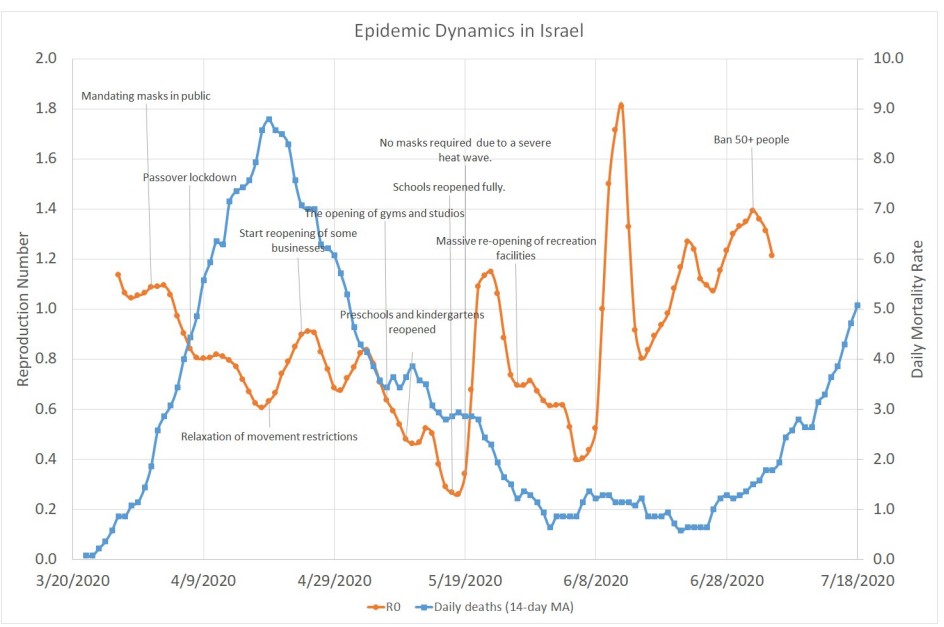

**Fig 1. Overall COVID–19 dynamics in Israel.** March 22–July 18, 2020.

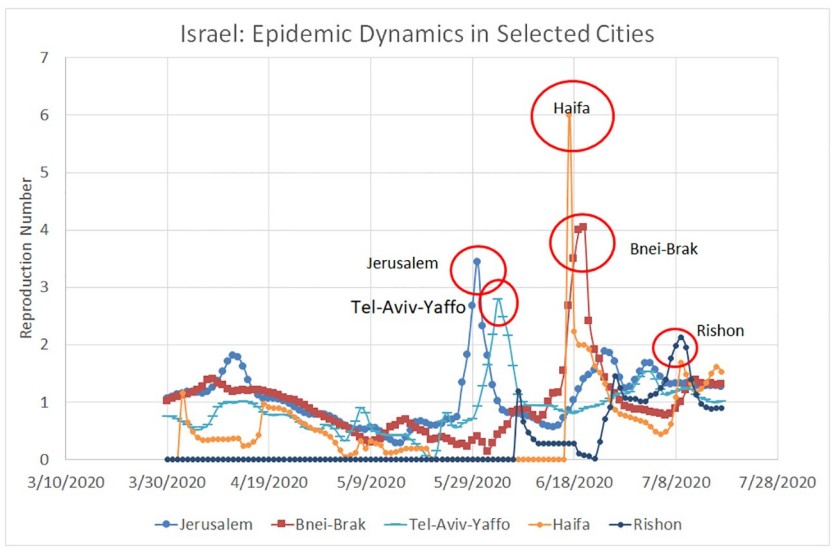

**Fig 2. Local COVID–19 dynamics in specific Israeli Cities.** March 30–July 17, 2020.

April 19, going down to 0.6 at the end of June, and rising up to 5.0 in the second half of July. However, Fig 1 does not reveal a consistent effect of any specific measures on the actual infection rate represented by the effective reproduction number.

Fig 2 shows the local evolution of the reproduction number in four most populous Israeli cities (Jerusalem, Tel-Aviv-Yaffo, Haifa, and Rishon-Le-Zion) along with the city of Bnei-Brak, which, for various reasons, has accumulated the second largest number of confirmed COVID-19 cases in Israel despite being only no. 9 in population size. Due to relatively low daily mortality rates in each city, $R_0$ was calculated over a 14-day sliding window of daily *confirmed* cases, smoothed with a five-day moving average. It is noteworthy that between the end of March and the end of May, the reproduction numbers in all five cities followed more or less the same decreasing trend. However, the months of June and July were characterised by distinct local peaks (brief but significant outbreaks), which can be easily related to fluctuations in $R_0$ at the country level that appear in Fig 1: May 23—an outbreak in Jerusalem and Tel-Aviv, June 12—an outbreak in Haifa and Bnei-Brak, June 22—another outbreak in Jerusalem, and July 2—an outbreak in Rishon-Le-Zion.

Fig 3 shows that, similar to Israel, the reproduction number in Greece quickly went from 1.2 down to 0.7, in parallel to domestic travel restrictions and various other lockdown measures. Despite several brief peaks in the infection rate, the overall decreasing trend continued throughout April, May, and June (when the lockdown restrictions were gradually removed), with $R_0$ staying well below the value of 1.0 most of the time and less than one death case per day on average.

As shown by Fig 4, the evolution of COVID-19 in Italy followed a pattern similar to Greece but on a completely different scale. The initial average value of the reproduction number in Italy was 2.5, about two times higher than in Israel and Greece, and it took it slightly more than one month to go down below 1.0, whereas the average daily number of deaths approached 800 in the beginning of April. Unlike in Israel, the lifting of lockdown restrictions in Italy was not followed by a significant increase in the value of $R_0$, which stayed below 1.0 from mid-March until the end of July. At the same time, the daily mortality rate in Italy faced a steady descent from 700 − 800 in March to nearly 10 in July. Fig 5 indicates that COVID-19

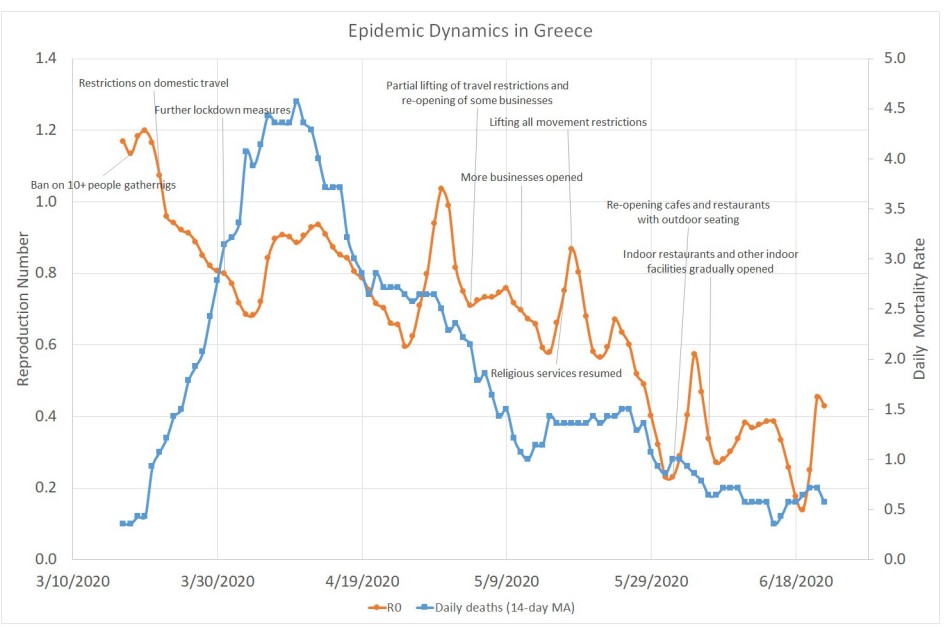

**Fig 3. COVID–19 dynamics in Greece.** March 17–June 22, 2020.

dynamics followed a very similar trend in all six Italian regions that had the highest number of death cases.

Fig 6 shows that the mean reproduction number in Sweden was at its peak value close to 2.0, much higher than its maximum values in Israel and Greece but lower than in Italy. Within the next two weeks it went down below 1.0, while the Swedish Government introduced various restrictions on social distancing rather than enforcing a total lockdown like in Israel, Italy,

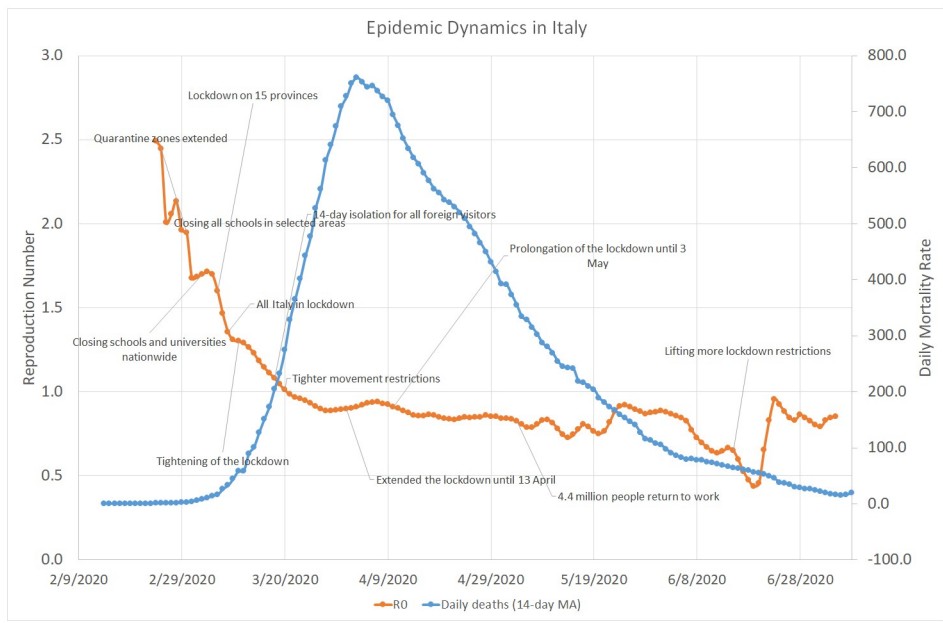

**Fig 4. COVID–19 dynamics in Italy.** February 14–July 09, 2020.

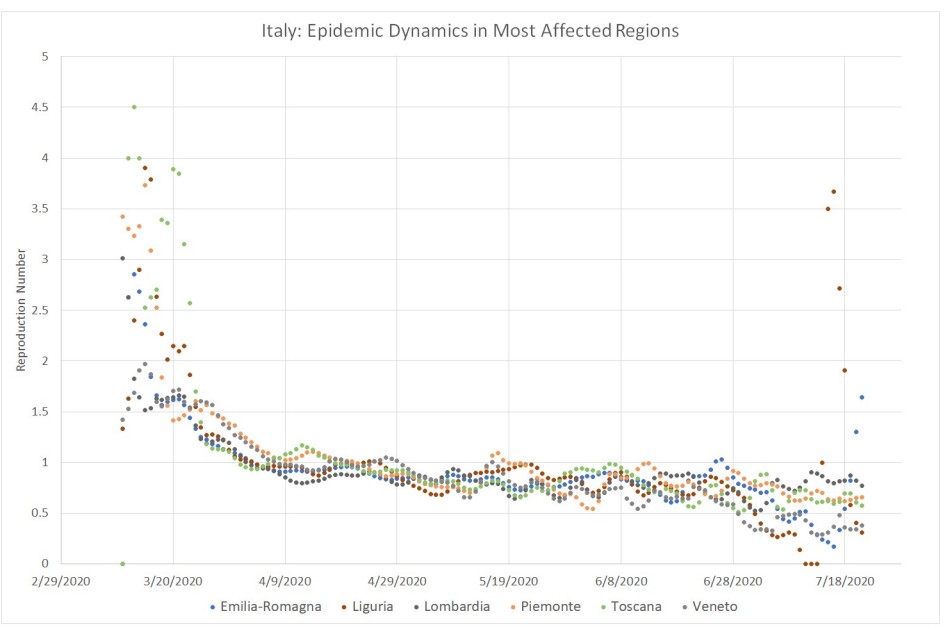

**Fig 5. Local COVID–19 dynamics in most affected Italian regions.** March 11–July 21, 2020.

Greece, and many other countries. While it is evident that Sweden did a better job than Israel in terms of "smoothing the curve", it experienced a relatively high mortality rate, with a peak number of about 100 cases per day at the end of April, which eventually went down below 15 in the second half of May.

Despite the differences in the demographic and cultural characteristics of the four countries, along with different intervention policies taken by their governments, all of them

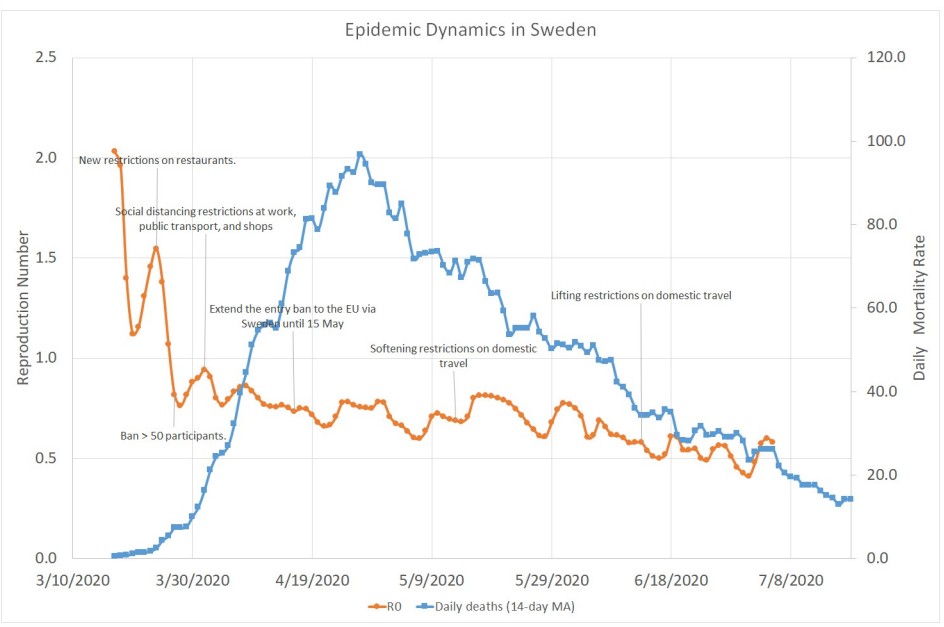

**Fig 6. COVID–19 dynamics in Sweden.** March 17–July 19, 2020.

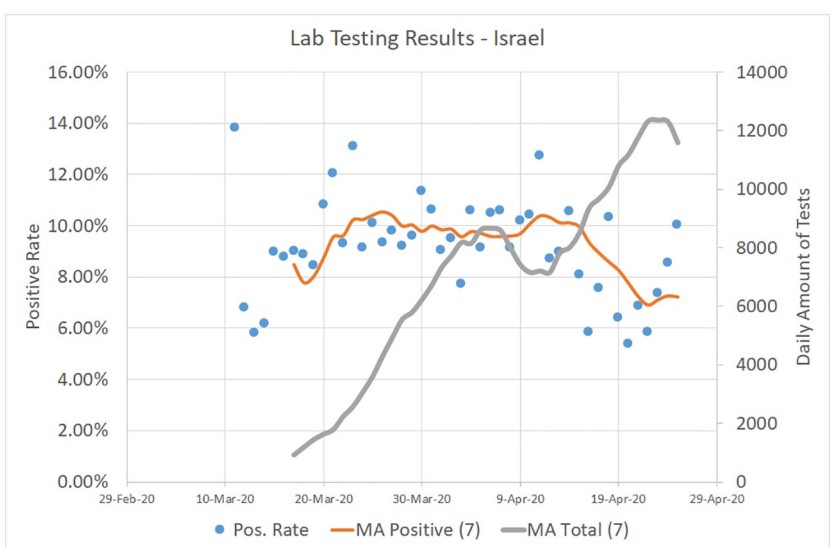

**Fig 7. Lab testing results in Israel.** March 11–April 25, 2020. Source: https://data.gov.il/dataset/covid-19.

experienced quite similar dynamics of COVID-19. The initial reproduction number varied between 1.2 and 2.5 and within the following few weeks, it rapidly decreased below 1.0, disregarding the specific social distancing measures introduced during that time. Subsequently, $R_0$ stayed below 1.0 for at least 1-2 months, without being significantly affected by various lockdown relief steps. A repeated increase in $R_0$ observed in Israel at a later stage requires further investigation, which may have implications for other countries as well.

The true Infection Rate (IR) of COVID-19, like any other epidemic, is a non–decreasing function of time. Hopefully, it will be estimated in the near future using massive serological tests but its true past values will remain unknown forever. However, we believe, like the authors of [14], that the Test Positivity Rate (TPR) during the first few weeks of the epidemic, calculated as the fraction of positive to total tests, should be an upper bound on the true IR. Fig 7 shows the 7–day moving averages for TPR and for the daily amount of COVID-19 tests performed in Israel between March 11 and April 25. Since the mean TPR reached its peak value of 10% around March 25, when the infection rate was also at its maximum, we can conclude that the true IR in Israeli population also did not exceed 10% at that time.

The results of initial 1,700 serological tests performed recently in Israel [15] suggest that the true infection rate in Israel is at least 10 times higher than the number of confirmed cases. Assuming that this reporting ratio is preserved and having 56, 000 confirmed cases up to July 22, we may estimate that by that date about 6.5% of the country's population of nearly 8.9 million have been infected with the virus. Using Eq 5 and the basic reproduction number of $R_0$ = 1.136, which was observed at the beginning of the epidemic, we can predict that the daily number of new cases will start a steady decline after the virus prevalence will exceed 12%.

A similar antibody test conducted between April 23 and June 3 on nearly 10, 000 residents of an Italian city of Bergamo has shown about 57% of positive results [16]. Considering the total amount of 13, 600 cases confirmed in the city of 122*k* people around the same time, the estimated reporting ratio in Bergamo stands at about 5. Using Eq 5 and the basic reproduction number of $R_0$ = 3.014, which was observed at the beginning of the epidemic in Lombardia Region, we can predict that the daily number of new cases in Bergamo will start a steady decline after the virus prevalence will exceed 67%, just slightly higher than 61% based on 14, 872 cases confirmed until July 22.

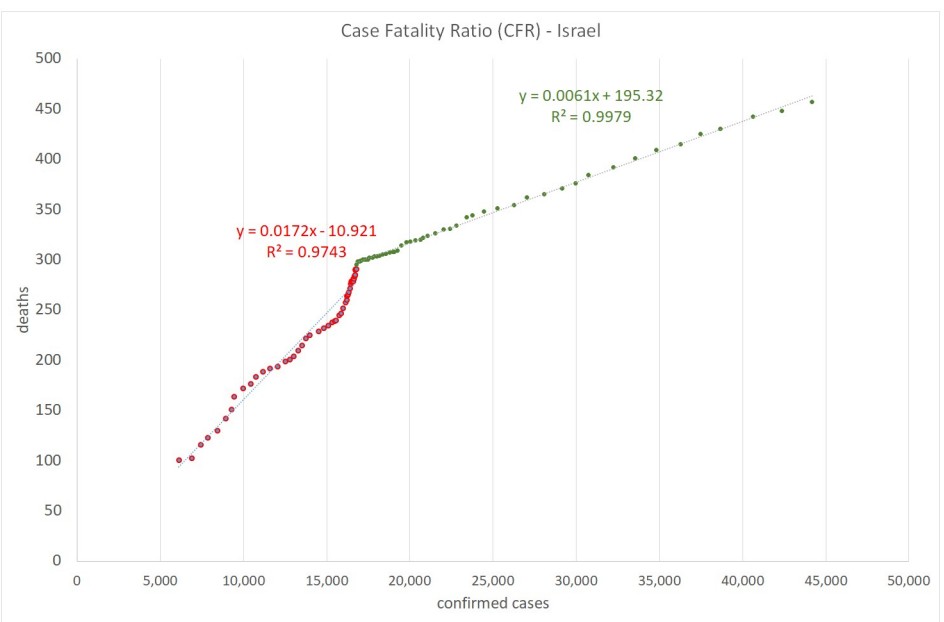

**Fig 8. Case fatality ratio in Israel.** Lag: 7 days.

Figs 8, 9, 10 and 11 show the calculation of the Case Fatality Ratio (CFR) in Israel, Greece, Italy, and Sweden, respectively, as a slope of a linear regression model between the cumulative amount of confirmed cases and the cumulative number of death outcomes 7 days later. However, in all four countries, we can observe a certain decrease in the line slope, i.e., in CFR, about 2-3 months from the start of the epidemic: from 1.8% to 0.5% in Israel, from 5.9% to 1.2% in Greece, from 16% to 13% in Italy, and from 14% to 2.9% in Sweden. The high values

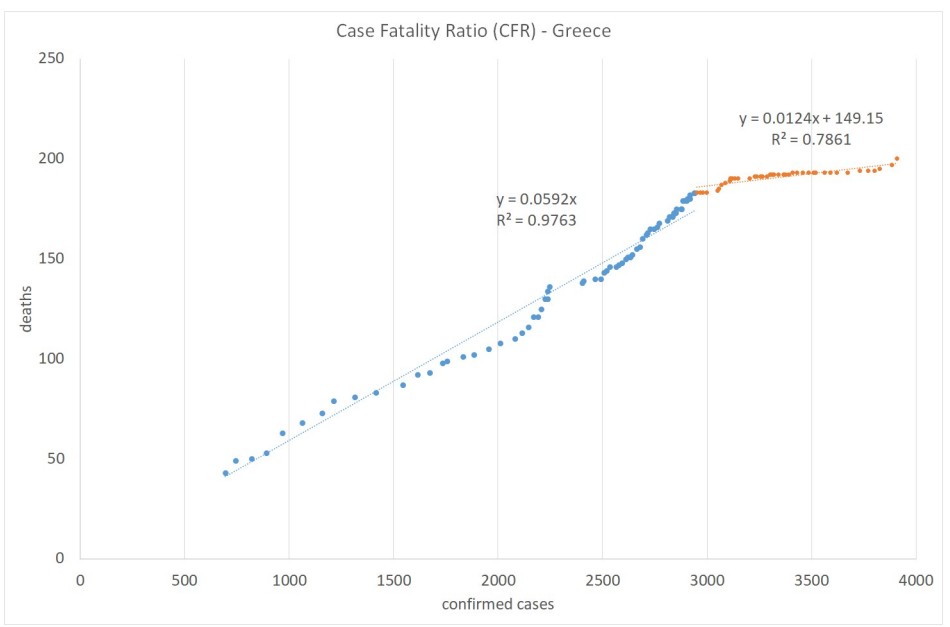

**Fig 9. Case fatality ratio in Greece.** Lag: 7 days.

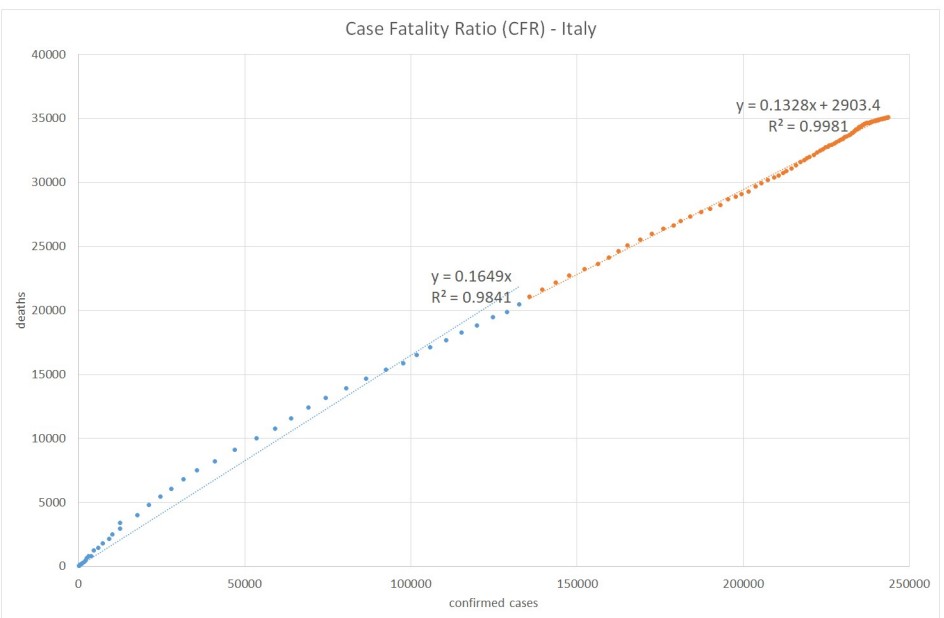

**Fig 10. Case fatality ratio in Italy.** Lag: 7 days.

of R-Square in all regression models indicate that most COVID-19 deaths are related to previously confirmed cases. This result supports the hypothesis that most symptomatic patients are detected by the lab tests. However, we do not have and may never have enough data to test this claim in either country. We also have no plausible explanation for the exceptionally high CFR values in Italy compared to Israel, Greece, and Sweden and to the world average of 3.6% in general. The reasons for the significant gaps between mortality rates in different countries and

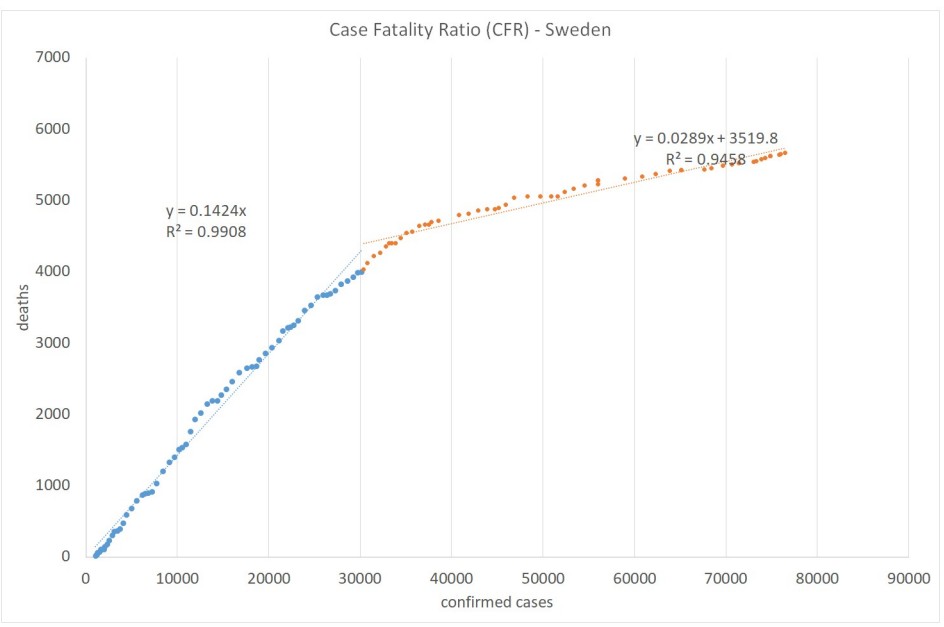

**Fig 11. Case fatality ratio in Sweden.** Lag: 7 days.

regions deserve a separate study as they may include differences in lab testing strategies, clinical treatment practices, outcome classification policies, coronavirus strains, etc.

## Projecting into the future

Based on the infection data estimated up to July 22, 2020 and assuming the reporting rate of $p = 10$ (according to the initial serology test conducted in Israel), Figs 12, 13 and 14 simulate the projected number of daily confirmed, cumulative confirmed, and cumulative death cases, respectively, for several possible values of the basic reproduction number $R_0$. Our simulation covers the period from the end of July to the end of December 2020, when we may be close to the peak of a new flu season in Israel. These projected numbers allow us to forecast three important parameters: maximum expected number of critical patients, total amount of death outcomes, and the start of a decline in the amount of new daily infections as a result of crossing the herd immunity threshold (calculated using Eq 5). Following [13], we assume that the average hospital stay of a critical patient is equal to 20 days and that the average proportion of critical patients in Israel is about 1% of confirmed cases. Consequently, the maximum number of available critical care beds should be equal to 20% of the maximum number of daily infections. The number of critical COVID-19 patients at the end of July 2020 is very close to the above estimator based on the moving average of new daily infections detected during the last two weeks of July. The total number of death outcomes is estimated as a product of the Case Fatality Ratio observed at the end of July (0.5%) by the cumulative number of cases confirmed 7 days earlier.

Here is a brief discussion of each simulated scenario:

- $R_0 = 1.00$. This value means that the daily infection rate has started a slow decline towards less than 100 cases by the end of October and less than 10 cases by the end of December, resulting in about 300 additional deaths. This scenario will require about 360 critical care beds at its peak, slightly higher than the actual number occupied at the end of July and much lower than the maximum capacity of 1, 000 beds.

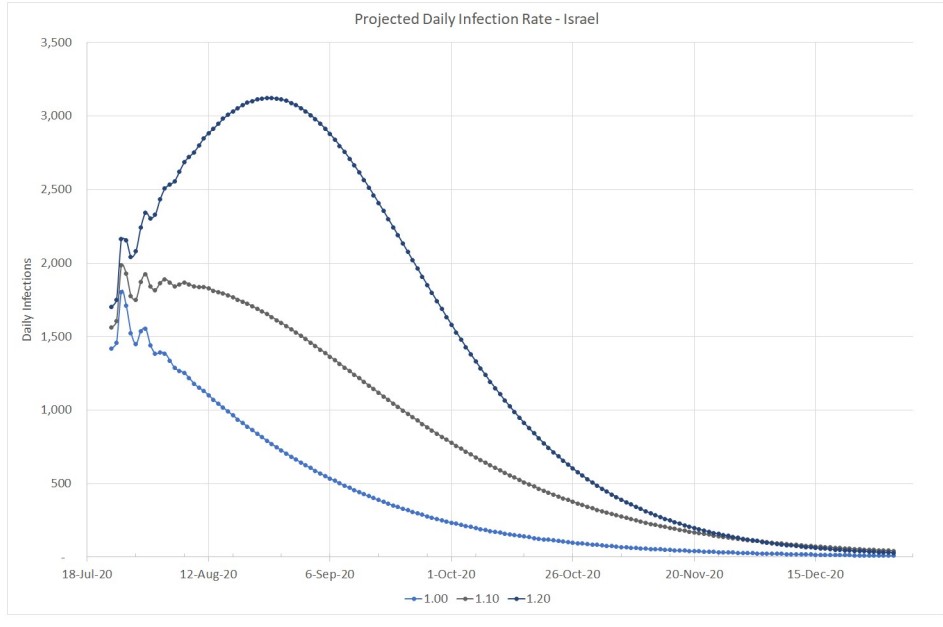

**Fig 12. Projected daily infection rate in Israel.** Up to December 31, 2020.

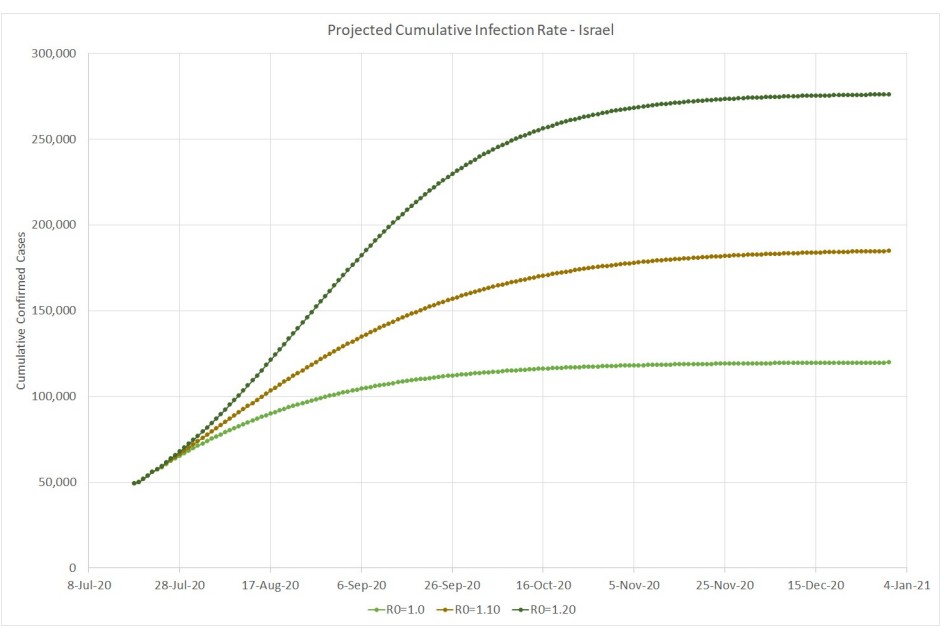

**Fig 13. Projected cumulative infection rate in Israel.** Up to December 31, 2020.

- $R_0$ = 1.10. This more severe scenario will increase the daily infection rate up to 1, 900 cases in the beginning of August, until, according to Eq 5, the amount of infected people in the Israeli population will reach the threshold of $(1 − 1/1.1) * 100 = 9.1\%$, equivalent to about 79, 000 confirmed cases. Then the infection rate will start a decrease down to 100 per day by the beginning of December. At its peak, this scenario will require close to 400 critical care beds. About 700 more people are expected to die from the virus until the end of 2020.

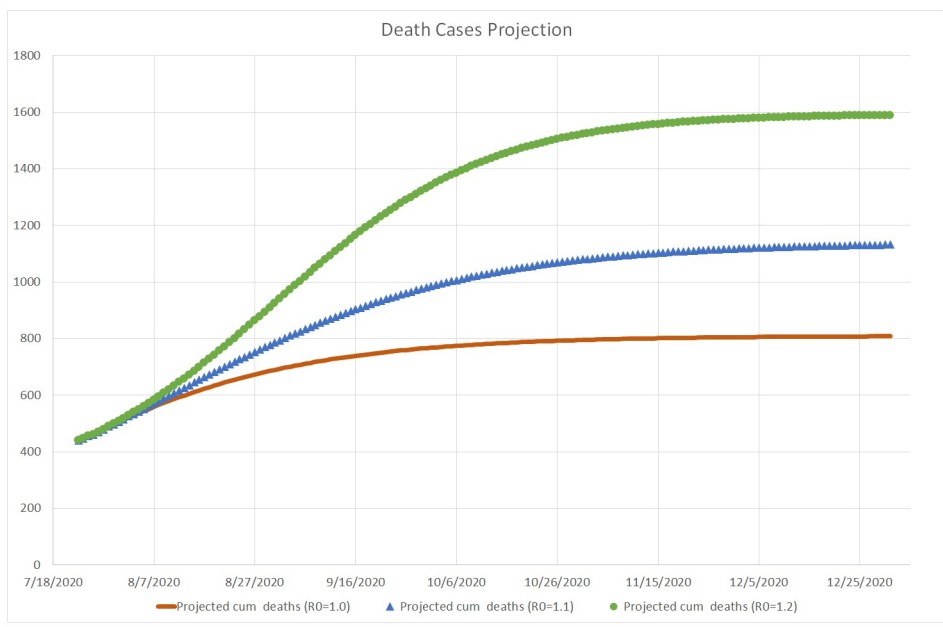

**Fig 14. Projected mortality rate in Israel.** Up to December 31, 2020.

- $R_0$ = 1.20. This may be the most realistic scenario, since as shown on Fig 1, similar values of the reproduction number were observed in Israel both at the beginning of the first wave, when virtually the entire population was susceptible, and during the first outbreak after the exit from the lockdown, when the population immunity was still very small. The peak is expected at the end of August, when the virus prevalence will reach $(1 - 1/1.2) * 100$ = 16.7%, equivalent to about 144, 000 confirmed cases. The corresponding daily infection rate of 3, 100 cases will require about 620 critical care beds, still well below the current capacity of the Israeli hospitals (1, 000 beds). Then the daily infection rate should go down to less than 100 cases by the beginning of December. Under this scenario, about 1, 100 more people are expected to die from the virus until the end of 2020.

## Conclusion

The future evolution of COVID-19 in Israel and elsewhere is hardly predictable. Until a vaccine becomes available, which may happen only by the end of 2020, the governments will have to strike the tough balance between health and economic issues. As shown by Eq 4, a decrease in the basic reproduction number $R_0$, e.g., as a result of a strict lockdown, should reduce the herd immunity threshold in terms of the total amount of infected people. However, the economic and psychological price of maintaining extreme social distancing measures until a massive vaccination program can be launched, may be too prohibitive. On the other hand, as shown by recent experience in Israel and some other countries, lifting a majority of restrictions creates a potential for multiple local outbreaks in places where $R_0$ exceeds 1/*fraction of susceptible population*. In the absence of an effective vaccine, the preparedness for such "second wave" phenomena is absolutely necessary.

A further progress in understanding the current pandemic will be possible with a release of detailed clinical records of COVID-19 patients in Israel and other countries to the research community. Analyzing these records with state–of—the–art statistical and machine learning algorithms may reveal answers to many important epidemiological questions such as an accurate and early detection of high–risk patients, identification of the most infectious persons ("super-spreaders") and locations, characterization of the most common infection pathways, etc. Many thousands of human lives worldwide are at risk and, as we all know, "data saves lives".

## Acknowledgments

I thank the Israeli Ministry of Health, the Gertner Institute, and the Johns Hopkins University Center for Systems Science and Engineering for publishing the COVID-19 data used in this research. I am also grateful to Michal Feldman, Sivan Frenkel, Neil Gandal, Isidore Last, Ady Pauzner, Avraham Tabbach, Matan Yonas (Tel-Aviv University), Yoram Louzoun (Bar-Ilan University), and Lev Muchnik (Hebrew University of Jerusalem) for the fruitful discussions of some of the topics presented in this paper.

## Author Contributions

**Conceptualization:** Mark Last.

**Data curation:** Mark Last.

**Formal analysis:** Mark Last.

**Investigation:** Mark Last.

**Methodology:** Mark Last.

**Software:** Mark Last.

**Validation:** Mark Last.

**Visualization:** Mark Last.

**Writing – original draft:** Mark Last.

**Writing – review & editing:** Mark Last.

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
