## [Decision Letter · Decision Letter 0]

10 Jun 2020

PONE-D-20-13386

The First Wave of COVID-19 in Israel - Initial Analysis of Publicly Available Data

PLOS ONE

Dear Dr. Last,

Thank you for submitting your manuscript to PLOS ONE. After careful consideration, we feel that it has merit but does not fully meet PLOS ONE’s publication criteria as it currently stands. Therefore, we invite you to submit a revised version of the manuscript that addresses the points raised during the review process.

The quantitative approach is too straightforward and should be seriously extended.

We look forward to receiving your revised manuscript.

Kind regards,

Stefan Cristian Gherghina, PhD. Habil.

Academic Editor

PLOS ONE

Journal Requirements:

Reviewers' comments:

Reviewer's Responses to Questions

**Comments to the Author**

1. Is the manuscript technically sound, and do the data support the conclusions?

Reviewer #1: Yes

Reviewer #2: Yes

2. Has the statistical analysis been performed appropriately and rigorously? 

Reviewer #1: I Don't Know

Reviewer #2: No

3. Have the authors made all data underlying the findings in their manuscript fully available?

Reviewer #1: Yes

Reviewer #2: Yes

4. Is the manuscript presented in an intelligible fashion and written in standard English?

Reviewer #1: Yes

Reviewer #2: Yes

5. Review Comments to the Author

Reviewer #1: While I found the article interesting and well developed, I personally believe it is too descriptive at this stage and very limited in the scope of analysis. While comparing Israel and Sweden because of similar size might sound compelling at first, more rigorous tests and comparisons are needed. The two countries have very different characteristics in terms of climate, culture, population density, rural vs. urban living, etc. - Hence, if a resubmission is intended, additional countries should be added.

Reviewer #2: This article studies the data available regarding COVID-19 infections and its fatality rate for Israel and Sweden. The aim of the study is to understand the effect of the implemented interventions and to foresee possible future scenarios.

It is an interesting study, but I have some comments on the descriptive analysis implemented.

1. The authors argue that they compare Israel and Sweden because they are countries of similar size. However, these two countries differ in many things that can affect on the evolution of the COVID19 contagion. As the author recognise, social distance is significantly different in both countries. However, there are many other substantial differences that can affect on the COVID19 dynamics such as the weather, the population density, the ability to telework, the general trust and the trust in government. It is important to discuss these substantial differences and its effect on the COVID19 dynamics.

2. The analysis would be much richer if they use data at some subnational level. It would be nice to see the analysis at the regional level. This will allow the authors also to control for some of the previous mentioned characteristics that can affect the COVI19 dynamics and that are different over the different regions in the country.

3. The linear regression that they do to estimate the correlation between confirmed cases and death should be improved. There are many characteristics that can affect in the relation of these two variables, such as the capacity of the health sector and the aging of the population. It would be very interesting if the authors implement the analysis at a regional level and departing from their initial regression, they include in a parsimonious way other control variables that can effect on the relation of these two variables. This would provide a better capacity to foresee the future scenarios depending on the regional characteristics.

4. In equation [1], the authors should replace R that refers to recovered people by another letter. It is misleading with R0

6. PLOS authors have the option to publish the peer review history of their article (what does this mean?). If published, this will include your full peer review and any attached files.

Reviewer #1: No

Reviewer #2: No

---

## [Author Response · Author response to Decision Letter 0]

2 Aug 2020

Response to Reviewers

PONE-D-20-13386

I am extremely grateful to the anonymous reviewers for their insightful comments and suggestions.

Reviewer #1: While I found the article interesting and well developed, I personally believe it is too descriptive at this stage and very limited in the scope of analysis. While comparing Israel and Sweden because of similar size might sound compelling at first, more rigorous tests and comparisons are needed. The two countries have very different characteristics in terms of climate, culture, population density, rural vs. urban living, etc. - Hence, if a resubmission is intended, additional countries should be added.

Response: Totally agree. The analysis now includes four countries: Israel, Greece, Italy, and Sweden.

Reviewer #2: This article studies the data available regarding COVID-19 infections and its fatality rate for Israel and Sweden. The aim of the study is to understand the effect of the implemented interventions and to foresee possible future scenarios.

It is an interesting study, but I have some comments on the descriptive analysis implemented.

1. The authors argue that they compare Israel and Sweden because they are countries of similar size. However, these two countries differ in many things that can affect on the evolution of the COVID19 contagion. As the author recognise, social distance is significantly different in both countries. However, there are many other substantial differences that can affect on the COVID19 dynamics such as the weather, the population density, the ability to telework, the general trust and the trust in government. It is important to discuss these substantial differences and its effect on the COVID19 dynamics.

Response: The differences between the four analyzed countries are now discussed in the Introduction and the Results sections.

2. The analysis would be much richer if they use data at some subnational level. It would be nice to see the analysis at the regional level. This will allow the authors also to control for some of the previous mentioned characteristics that can affect the COVI19 dynamics and that are different over the different regions in the country.

Response: An excellent idea! The paper now includes regional level analysis for Israel (see Fig. 2) and Italy (see Fig. 5).

3. The linear regression that they do to estimate the correlation between confirmed cases and death should be improved. There are many characteristics that can affect in the relation of these two variables, such as the capacity of the health sector and the aging of the population. It would be very interesting if the authors implement the analysis at a regional level and departing from their initial regression, they include in a parsimonious way other control variables that can effect on the relation of these two variables. This would provide a better capacity to foresee the future scenarios depending on the regional characteristics.

Response: The correlation between confirmed cases and deaths is now explored with respect to the time dimension. Unfortunately, the effect of additional important characteristics suggested by the reviewer could not be analyzed due to the unavailability of sufficiently detailed data on the countries included in this study.

4. In equation [1], the authors should replace R that refers to recovered people by another letter. It is misleading with R0

Response: In Eq. 1, R was replaced with H, for Healed, to avoid confusion with R0.

---

## [Decision Letter · Decision Letter 1]

7 Sep 2020

PONE-D-20-13386R1

The First Wave of COVID-19 in Israel - Initial Analysis of Publicly Available Data

PLOS ONE

Dear Dr. Last,

Thank you for submitting your manuscript to PLOS ONE. After careful consideration, we feel that it has merit but does not fully meet PLOS ONE’s publication criteria as it currently stands. Therefore, we invite you to submit a revised version of the manuscript that addresses the points raised during the review process.

The manuscript requires a couple of further explanations, whereas the literature review should be extended.

We look forward to receiving your revised manuscript.

Kind regards,

Stefan Cristian Gherghina, PhD. Habil.

Academic Editor

PLOS ONE

Reviewers' comments:

Reviewer's Responses to Questions

**Comments to the Author**

1. If the authors have adequately addressed your comments raised in a previous round of review and you feel that this manuscript is now acceptable for publication, you may indicate that here to bypass the “Comments to the Author” section, enter your conflict of interest statement in the “Confidential to Editor” section, and submit your "Accept" recommendation.

Reviewer #3: All comments have been addressed

Reviewer #4: All comments have been addressed

Reviewer #5: All comments have been addressed

2. Is the manuscript technically sound, and do the data support the conclusions?

Reviewer #3: Partly

Reviewer #4: Yes

Reviewer #5: Yes

3. Has the statistical analysis been performed appropriately and rigorously? 

Reviewer #3: Yes

Reviewer #4: Yes

Reviewer #5: Yes

4. Have the authors made all data underlying the findings in their manuscript fully available?

Reviewer #3: Yes

Reviewer #4: Yes

Reviewer #5: Yes

5. Is the manuscript presented in an intelligible fashion and written in standard English?

Reviewer #3: Yes

Reviewer #4: Yes

Reviewer #5: Yes

6. Review Comments to the Author

Reviewer #3: I read it with interest and enjoyed the clarity and brevity of your analysis. Nonetheless, I have decided not to accept immediately. While I found your analysis and literature to be interesting, the study would resonate more among our readers if it offered insights into the current literature by acknowledging these followings papers, which were focused on the Vietnam, lauded as the coronavirus fighting in the first wave. I believe that these measures as well as the study would have merits for the authors to enhance their current literature:

[a] Huynh, T. L. D. (2020). Does culture matter social distancing under the COVID-19 pandemic?. Safety Science, 104872.

[b] Huynh, T. L. D. (2020). “If You Wear a Mask, Then You Must Know How to Use It and Dispose of It Properly!”: A Survey Study in Vietnam. Review of Behavioral Economics, 7(2), 145-158.

[c] Huynh, T. L. D. (2020). The COVID-19 containment in Vietnam: What are we doing?. Journal of Global Health, 10(1).

[d] Huynh, T. L. (2020). The COVID-19 risk perception: A survey on socioeconomics and media attention. Econ. Bull, 40(1), 758-764.

[e] Huynh, T. L. D. (2020). Data for understanding the risk perception of COVID-19 from Vietnamese sample. Data in Brief, 105530.

After that, this manuscript is ready to be published.

Reviewer #4: Related to the 4th point by Reviewer 2, there are still a couple of places "R0" was written, for example p2, p12, p13. Please check and correct.

Reviewer #5: This article explores the evolution of Covid-19 in Israel compared to Greece, Italy and Sweden, and aims to understand the effect of the implemented interventions and to predict possible future scenarios. Even though it is mostly at descriptive level, concerning that every single piece of analysis adds to the information set of humanity, it might be valuable to publish it, once the necessary implementations are made.

It seems that the author successfully implemented most of the recommendations of the referees. He added two more countries, Greece and Italy, and two subnational analysis, one for Israel and one for Italy. Additionally, the graphs and the regression analyses are clearer and that`s why make more sense compared to the initial submission. Thus, I think the paper has improved a lot.

I would recommend couple of minor additions:

1. The “Current mortality rate” needs an explanation of what that current date is. Plus a “current number of confirmed cases per 100K” would be added to Table 1 to give a clearer picture.

2. The y-axis of Figure 2 is not stated. The author should add a title to that axis.

3. I do not think the author needs to spend 3-4 sentences per figure to explain everything in the Figures 3-4-5. They are already very clear. He can safely remove some. This will shorten the paper a little.

Once these minor additions are added, I think the paper will add to the literature and can be published.

7. PLOS authors have the option to publish the peer review history of their article (what does this mean?). If published, this will include your full peer review and any attached files.

Reviewer #3: No

Reviewer #4: No

Reviewer #5: No

---

## [Author Response · Author response to Decision Letter 1]

23 Sep 2020

Response to Reviewers

PONE-D-20-13386R1

I am again extremely grateful to the anonymous reviewers for their additional comments and suggestions.

Reviewer #3: I read it with interest and enjoyed the clarity and brevity of your analysis. Nonetheless, I have decided not to accept immediately. While I found your analysis and literature to be interesting, the study would resonate more among our readers if it offered insights into the current literature by acknowledging these followings papers, which were focused on the Vietnam, lauded as the coronavirus fighting in the first wave. I believe that these measures as well as the study would have merits for the authors to enhance their current literature:

[a] Huynh, T. L. D. (2020). Does culture matter social distancing under the COVID-19 pandemic?. Safety Science, 104872.

[b] Huynh, T. L. D. (2020). “If You Wear a Mask, Then You Must Know How to Use It and Dispose of It Properly!”: A Survey Study in Vietnam. Review of Behavioral Economics, 7(2), 145-158.

[c] Huynh, T. L. D. (2020). The COVID-19 containment in Vietnam: What are we doing?. Journal of Global Health, 10(1).

[d] Huynh, T. L. (2020). The COVID-19 risk perception: A survey on socioeconomics and media attention. Econ. Bull, 40(1), 758-764.

[e] Huynh, T. L. D. (2020). Data for understanding the risk perception of COVID-19 from Vietnamese sample. Data in Brief, 105530.

After that, this manuscript is ready to be published.

Response: I found the above references very useful and extended the literature survey in the Introduction section accordingly.

Reviewer #4: Related to the 4th point by Reviewer 2, there are still a couple of places "R0" was written, for example p2, p12, p13. Please check and correct.

Response: checked and corrected as necessary.

Reviewer #5: This article explores the evolution of Covid-19 in Israel compared to Greece, Italy and Sweden, and aims to understand the effect of the implemented interventions and to predict possible future scenarios. Even though it is mostly at descriptive level, concerning that every single piece of analysis adds to the information set of humanity, it might be valuable to publish it, once the necessary implementations are made.

It seems that the author successfully implemented most of the recommendations of the referees. He added two more countries, Greece and Italy, and two subnational analysis, one for Israel and one for Italy. Additionally, the graphs and the regression analyses are clearer and that`s why make more sense compared to the initial submission. Thus, I think the paper has improved a lot.

I would recommend couple of minor additions:

1. The “Current mortality rate” needs an explanation of what that current date is. Plus a “current number of confirmed cases per 100K” would be added to Table 1 to give a clearer picture.

Response: The word “current” was replaced with specific dates, where applicable (including Table 1). The number of confirmed cases was added to Table 1 as well.

2. The y-axis of Figure 2 is not stated. The author should add a title to that axis.

Response: The y-axis title added.

3. I do not think the author needs to spend 3-4 sentences per figure to explain everything in the Figures 3-4-5. They are already very clear. He can safely remove some. This will shorten the paper a little.

Response: Several sentences removed.

Once these minor additions are added, I think the paper will add to the literature and can be published

---

## [Decision Letter · Decision Letter 2]

28 Sep 2020

The First Wave of COVID-19 in Israel - Initial Analysis of Publicly Available Data

PONE-D-20-13386R2

Dear Dr. Last,

We’re pleased to inform you that your manuscript has been judged scientifically suitable for publication and will be formally accepted for publication once it meets all outstanding technical requirements.

Kind regards,

Stefan Cristian Gherghina, PhD. Habil.

Academic Editor

PLOS ONE

Additional Editor Comments (optional):

Reviewers' comments:

Reviewer's Responses to Questions

**Comments to the Author**

1. If the authors have adequately addressed your comments raised in a previous round of review and you feel that this manuscript is now acceptable for publication, you may indicate that here to bypass the “Comments to the Author” section, enter your conflict of interest statement in the “Confidential to Editor” section, and submit your "Accept" recommendation.

Reviewer #3: All comments have been addressed

Reviewer #5: All comments have been addressed

2. Is the manuscript technically sound, and do the data support the conclusions?

Reviewer #3: Yes

Reviewer #5: Yes

3. Has the statistical analysis been performed appropriately and rigorously? 

Reviewer #3: Yes

Reviewer #5: Yes

4. Have the authors made all data underlying the findings in their manuscript fully available?

Reviewer #3: Yes

Reviewer #5: Yes

5. Is the manuscript presented in an intelligible fashion and written in standard English?

Reviewer #3: Yes

Reviewer #5: Yes

6. Review Comments to the Author

Reviewer #3: This revised version incorporated all my comments. Therefore, I am quite optimistic about the publishing opportunity from this manuscript.

Reviewer #5: This article aims to address the effect of the implemented interventions across various countries to predict possible future scenarios. The authors successfully implemented all of my recommendations and most of the others referees` too.

Well-done. Congratulations

7. PLOS authors have the option to publish the peer review history of their article (what does this mean?). If published, this will include your full peer review and any attached files.

Reviewer #3: **Yes: **Toan Luu Duc Huynh

Reviewer #5: **Yes: **Orhan Erdem

---

## [Editor Report · Acceptance letter]

5 Oct 2020

PONE-D-20-13386R2 

The First Wave of COVID-19 in Israel – Initial Analysis of Publicly Available Data 

Dear Dr. Last:

I'm pleased to inform you that your manuscript has been deemed suitable for publication in PLOS ONE. Congratulations! Your manuscript is now with our production department. 

Kind regards, 

on behalf of

Dr. Stefan Cristian Gherghina 

Academic Editor

PLOS ONE